



**Zeppelin-led study on the onset of new particle formation in the planetary boundary layer**
Janne Lampilahti[1], Hanna E. Manninen[2], Tuomo Nieminen[1], Sander Mirme[3], Mikael Ehn[1], Iida
Pullinen[4], Katri Leino[1], Siegfried Schobesberger[1,4], Juha Kangasluoma[1], Jenni Kontkanen[1], Emma
Järvinen[5], Riikka Väänänen[1], Taina Yli-Juuti[4], Radovan Krejci[6], Katrianne Lehtipalo[1,7], Janne
Levula[1], Aadu Mirme[3], Stefano Decesari[8], Ralf Tillmann[9], Douglas R. Worsnop[1,4,10], Franz Rohrer[9],
Astrid Kiendler-Scharr[9], Tuukka Petäjä[1,11], Veli-Matti Kerminen[1], Thomas F. Mentel[9], and Markku
Kulmala[1,11,12]
[1]Institute for Atmospheric and Earth System Research / Physics, Faculty of Science, University of
Helsinki, Helsinki, Finland.
[2]CERN, CH-1211 Geneva, Switzerland.
[3]Institute of Physics, University of Tartu, Tartu, Estonia.
[4]Department of Applied Physics, University of Eastern Finland, Kuopio, Finland.
[5]National Center for Atmospheric Research, Boulder, CO, USA.
[6]Department of Environmental Science & Bolin Centre for Climate research, Stockholm University,
Stockholm, Sweden.
[7]Finnish Meteorological Institute, Helsinki, Finland.
[8]Istituto di Scienze dell'Atmosfera e del Clima, CNR, Bologna, Italy.
[9]Institute for Energy and Climate Research, IEK-8, Forschungszentrum Jülich GmbH, Jülich,
Germany.
[10]Aerodyne Research Inc, Billerica, MA, USA.
[11]Joint International Research Laboratory of Atmospheric and Earth System Sciences, Nanjing.
University, Nanjing, China.
[12]Aerosol and Haze Laboratory, Beijing Advanced Innovation Center for Soft Matter Science and
Engineering, Beijing University of Chemical Technology, Beijing, China.
Correspondence to: Janne Lampilahti (janne.lampilahti@helsinki.fi)
**Abstract**
We compared observations of aerosol particle formation and growth in different parts of the
planetary boundary layer at two different environments that have frequent new particle formation
(NPF) events. In summer 2012 we had a campaign in Po Valley, Italy (urban background) and in
spring 2013 a similar campaign took place in Hyytiälä, Finland (rural background). Our study



consists of airborne and ground-based measurements of ion and particle size distribution from ~1
nm. The airborne measurements were performed using a Zeppelin inside the boundary layer up to
1000 m altitude. Our observations show the onset of regional NPF and the subsequent growth of the
aerosol particles happening uniformly inside the mixed layer (ML) in both locations. However, in
Hyytiälä we noticed local enhancement in the intensity of NPF caused by mesoscale BL dynamics.
Additionally, our observations indicate that in Hyytiälä NPF was probably also taking place above
the ML. In Po Valley we observed NPF that was limited to a specific air mass.

**43 1 Introduction**

The boundary layer (BL) is the lowest layer of the earth's atmosphere (Stull, 1988). The BL is an
interface controlling the exchange of mass and energy between atmosphere and surface. Ground
based measurements are often used as representative observations for the whole BL. However they
cannot cover vertical internal variability of BL and this can be addressed only by airborne
observations.

Figure 1 show the typical BL evolution over land during the time span on one day. Shortly after
sunrise convective mixing creates a mixed layer (ML) that rapidly grows during the morning by
entraining air from above and can reach an altitude of ~1-2 km above the surface. The ML is capped
by a stable layer at the top. Above the BL is the free troposphere (FT), which is decoupled from the
surface. Here we define BL to mean all the layers below the FT. Around sunset convective mixing
and turbulence diminishes and the ML becomes what is known as the residual layer (RL). During
the night a stable boundary layer develops due to interaction with the ground surface. This layer has
only weak intermittent turbulence and it smoothly blends into the RL.

We studied where new particle formation (NPF) occurs in the BL and how it relates to BL
evolution, comparing two different environments. NPF refers to the formation of nanometer sized
clusters from low-volatility vapors present in the atmosphere, and their subsequent growth to larger
aerosol particles (Kulmala et al., 2013). Understanding NPF better is of major interest, since it is a
dominant source of cloud condensation nuclei in the atmosphere and therefore can have important
indirect effects on climate (Dunne et al., 2016; Gordon et al., 2017; Pierce and Adams, 2009; Yu and
Luo, 2009).

Nilsson et al. (2001) studied NPF in a boreal forest environment and observed that in addition to
increased solar radiation the onset of turbulence appears to be a necessary trigger for NPF. Several



explanations for this connection were proposed: NPF might be starting in the RL or at the top of the
shallow ML, from where the aerosol particles are mixed to the surface as the ML starts to grow.
NPF starts in the ML due to dilution of pre-existing aerosol and drop in vapor sink. Convective
mixing brings different precursor gases, one present in the RL and the other in the ML, into contact
with each other initiating NPF inside the ML.

Airborne measurements of nanoparticles from different environments show that NPF occurs in
many parts of the BL. Multiple observations from Central Europe suggest that aerosol particles are
formed on top of a shallow ML (Platis et al., 2015; Siebert et al., 2004; Chen et al., 2018) or inside
the RL (Stratmann et al., 2003; Wehner et al., 2010). Other results come from a boreal forest
environment in southern Finland. Lampilahti et al. (2020a) showed evidence that NPF may occur in
the interface between the RL and the FT. O'Dowd et al. (2009) observed the first signs of NPF in
the surface ML and Leino et al. (2019) showed that sub-3 nm particles have higher concentration
close to surface. Laakso et al. (2007) performed hot-air balloon measurements and concluded that
NPF either took place throughout the ML or in the lower part of the ML. Measurements by
Schobesberger et al. (2013) suggested that NPF was more intense in the top parts of a developed
ML. More measurements are needed in order to understand these mixed results.

Here we present NPF measurements on board a Zeppelin airship carried out during the EU
supported PEGASOS (Pan-European Gas-AeroSOls Climate Interaction Study) project. The main
goal of the project was to quantify the magnitude of regional to global feedbacks between
atmospheric chemistry and physics, and thus quantify their impact on the changing climate. The
Zeppelin flights were used to observe radicals, trace gases, and aerosol particles inside the lower
troposphere over Europe in several locations during 2012-2013.

By using a Zeppelin NT (Neue Technologie) airship we were able to sample from a stable, agile
platform, up to 1000 meters above sea level (asl). The high payload capacity of the Zeppelin
enabled us to carry state-of-the-art instrumentation, specifically designed to collect information on
the feedback processes between the chemical compounds and the smallest aerosol particles to better
estimate their role in climate and air quality.

The NPF focused campaigns presented here were performed in Po Valley, Italy, and Hyytiälä,
Finland. At both locations NPF events happen frequently. Po Valley represents urban background
conditions where anthropogenic emissions are an important source of gaseous precursors for NPF



(e.g. Kontkanen et al., 2016). Hyytiälä represents rural background conditions where organic vapors
from the surrounding forests play a major role in NPF (e.g. Dada et al., 2017).

Here we combine comprehensive ground-based and airborne measurements to compare two NPF
cases from Po Valley to one case from Hyytiälä. The Zeppelin allowed us to repeatedly profile the
lowest 1 km of the atmosphere providing a full picture of what is happening in the BL during the
onset of NPF. We will show in which part or parts of the BL the onset of NPF and the subsequent
particle growth occurred at the two measurement sites as well as determine formation and growth
rates for the aerosol particles.

**2 Methods**
San Pietro Capofiume in Po Valley, Italy and Hyytiälä in Southern Finland are interesting
environments to compare from nucleation and particle growth point of view because NPF is
frequently observed in both environments. The vertical measurement profiles analyzed in this study
were performed in a close proximity to the ground-based measurement sites.

**2.1 San Pietro Capofiume, Italy**
San Pietro Capofiume (SPC, 44°39'N 11°37'E, 11 m asl) is located in the eastern part of Po Valley,
Italy, between the cities of Bologna and Ferrara. Po Valley is considered a pollution hot spot,
although, the station itself is surrounded by vast agricultural fields away from point sources. Thus
the aerosol concentration and composition at SPC reflect the Po Valley regional background. NPF is
frequently observed in SPC (36% of days) with maxima in May and July (Hamed et al., 2007;
Laaksonen et al., 2005).

The instruments measuring the aerosol particle number-size distribution were a scanning mobility
particle sizer (SMPS, 10-700 nm, 5 min time resolution; Wiedensohler et al., 2012) and a neutral
cluster and air ion spectrometer (NAIS, particles: ~2-40 nm, ions: 0.8-40 nm, 4 min time resolution;
Mirme and Mirme, 2013). We used the NAIS's positive polarity for the particle number size
distribution data. The ML height was determined from ceilometer (Lufft CHM 15k) measurements.
Basic meteorology and $SO_2$ gas concentration data (Thermo 43iTLE monitor) were also available at
surface level (2-3 m above ground level).

**2.2 Hyytiälä, Finland**



In Finland the ground-based measurements were performed at the SMEAR II (Station for
Measuring Forest Ecosystem-Atmosphere Relations II) station located in Hyytiälä, Finland (HTL,
61°51'N 24°17'E, 181 m asl; Hari and Kulmala, 2005). The station is equipped with extensive
facilities to measure the forest ecosystem and the atmosphere. The measurement site is surrounded
by coniferous boreal forest.

The forest emits biogenic volatile organic compounds (Hakola et al., 2003), which can be oxidized
in the atmosphere to form low-volatile vapors that contribute to aerosol particle formation and
growth (Ehn et al., 2014; Mohr et al., 2019). NPF is frequently observed in HTL (23% of all days),
especially in spring and autumn (Dal Maso et al., 2005; Nieminen et al., 2014).
Aerosol particle and ion number-size distributions were measured by the station's differential
mobility particle sizer (DMPS, 3-1000 nm, 10 min time resolution; Aalto et al., 2001) and the NAIS
(Manninen et al., 2009). Sub-3 nm particle number-size distribution was measured by a particle size
magnifier running in scanning mode (PSM, 1.2-2.5 nm, 10 min time resolution; Vanhanen et al.,
2011). Also a PSM measured at SPC but we were not able to reliably calculate formation rates from
the data. Basic meteorological variables, radiation, and $SO_2$ were measured from the station's mast
at 16.8 meters above ground. In addition, a supporting NPF forecast tool was developed to aid the
planning of research flights (Nieminen et al., 2015).

**2.3 Zeppelin NT airship**
A Zeppelin NT airship was used for monitoring the atmosphere below 1 km. The aerosol particles
and trace gases were sampled with instrumentation installed inside the Zeppelin's cabin. The
Zeppelin operated with three different instrument layouts. A specific layout was chosen according to
the flight plan and scientific aim of the flight.

Here we analyzed data from measurement flights that carried the so-called nucleation layout.
Instruments specific to this layout were the atmospheric pressure interface time-of-flight mass
spectrometer (APi-TOF; Junninen et al., 2010), used for measuring the elemental composition of
naturally charged ions and the NAIS for particle and ion number size distributions. We also used the
aerosol number-size distribution data from the SMPS (10-400 nm, 4 min time resolution) and PSM
running in scanning mode, which were on board during all the measurement flights. The size range
and time resolution of the onboard NAIS and PSM were same as for the instruments in HTL (see
Section 2.1).



During a measurement flight the Zeppelin did multiple vertical profiles over a small area (~10 km$^2$).
The profiling spot was picked typically down-wind from the measurement site in order not to
compromise the ground-based measurements with any emissions. The vertical extent of the profiles
was ~100-1000 m above the ground. The airspeed during measurement was ~20 m/s and the
vertical speed during ascend and descend was ~0.5 m/s and ~3 m/s respectively.

**2.4. Cessna 172 airplane**
During the PEGASOS northern mission in spring 2013, a Cessna 172 airplane carrying scientific
instrumentation was deployed to measure aerosol particles, trace gases and meteorological variables
in the lower troposphere alongside the Zeppelin. The measurement setup and instrumentation on
board have been described in previous studies (Schobesberger et al., 2013; Lampilahti et al., 2020c;
Leino et al., 2019; Väänänen et al., 2016).

Basic meteorological variables (temperature, pressure, relative humidity) were measured on board.
Particle number-size distribution was measured using a SMPS (10-400 nm size range, 2 min time
resolution) and the number concentration of >3 nm particles was measured using an ultrafine
condensation particle counter (UF-CPC, TSI model 3776) at 1 s time resolution. The altitude range
of the airplane was ~100-3000 m above ground and the measurement airspeed was 36 m/s.

**2.5 Flight profiles and atmospheric conditions**
Our measurements focused on the time of BL development from sunrise until noon (Figure 1). This
is the time when the onset of NPF is typically observed at the ground level. The vertical profile
measurements represent the particle and gas concentrations in the lower parts of the atmosphere: the
mixed layer, the residual layer, the nocturnal boundary layer. At the same time, the ground-based
measurements recorded conditions in the surface layer. Here we consider the BL to include all the
atmospheric layers below the free troposphere.

The basic conditions for the Zeppelin flights in both Italy and Finland were clear sky and low wind
speed. Under these conditions, the sun heats the surface during the morning, which drives intense
vertical mixing.

**2.6 Data analysis**
The onset of NPF occurs when low-volatility vapors in the atmosphere form nanometer sized
clusters that continue to grow to larger aerosol particles (Kulmala et al., 2013).




We determined the onset of a NPF event visually from the initial increase in the number
concentration of intermediate (2-4 nm) air ions at the beginning of the NPF event. An increase in
the intermediate ion concentration has been identified as a good indicator for NPF (Leino et al.,
2016).  This is because an increase in the number concentration of intermediate ions is usually due
to NPF and otherwise the number concentration is extremely low (below 5 cm$^{-3}$).

Particle growth rates (GR), formation rates and coagulation sinks were calculated in different size
ranges according to the methods described by Kulmala et al. (2012). For particles and ions in the 1-
2 nm and 2-3 nm size range the GR was determined from the ion number-size distribution measured
by the NAIS. During NPF the number concentration in each size channel increased sequentially as
the freshly formed particles grew larger. We determined the time when the number concentration
began to rise in each size bin by fitting a sigmoid function to the rising concentration edge and
finding the point where the sigmoid reached 75% of its maximum value (appearance time method;
Lehtipalo et al., 2014). The corresponding diameter in each size bin was the bin's geometric mean
diameter. Before the fitting procedure the number concentrations were averaged using a 15 min
median and after that divided by the maximum concentration value in each size channel.

For larger particles and ions (3-7 nm and 7-20 nm) the GR was determined by fitting a log-normal
distribution over the growing nucleation mode at each time step and assigning the fitted curve's
peak value as the corresponding mode diameter. In each size range a value for the GR was obtained
as the slope of a linear least squares fit to the time-diameter value pairs.

The formation rate of 1.5 nm particles and ions was determined from the PSM data and the NAIS
ion data respectively (Kulmala et al., 2012). The formation rate of 3 nm particles and ions was
determined from the NAIS data. Coagulation sinks were calculated from the SMPS or DMPS data.
Condensation sink for sulfuric acid was calculated from the Zeppelin's on board SMPS.

Sulfuric acid (SA) is a key compound in atmospheric nucleation (Sipilä et al., 2010). As we did not
have direct measurements of SA concentration, we used [HSO4-] from the APi-TOF measurements
as a qualitative indicator of [H2SO4] and named it pseudo-SA. To determine this pseudo-SA, we
summed up all ions containing HSO4-, e.g. the ion itself but also larger clusters, like
(H2SO4)$_n$*HSO4-. We assumed steady state conditions and that the concentration of SA-containing
ions is much lower than the total ion concentration. Under these conditions [HSO4-] (including all





clusters where this ion was present) can be considered close to a linear function of [H2SO4] (Eisele
and Tanner, 1991). At the highest SA loadings, ions with HSO4- can be a dominant fraction of the
total ions (Ehn et al., 2010), in which case the linearity no longer holds. In addition, this assumes a
constant concentration of ions, although for example the sinks for ions can vary, e.g. by an
increased particle concentration. As such, the pseudo-SA parameter should indeed only be
considered a qualitative indicator for SA.

In SPC the ML height was derived from the ceilometer measurements. However, in HTL weak
scattering signal prevented reliable determination of ML height using the on-site lidar. For this
reason in HTL the ML height was determined from vertical profiles of meteorological variables and
aerosol particle concentrations on board the Zeppelin and the Cessna 172 airplane. In these profiles
the top of the ML was revealed by the maximum positive gradient in potential temperature and
minimum negative gradient in humidity and total particle number concentration (Stull, 1988).

The origin of the air masses was investigated using back trajectory analysis. The trajectories were
calculated with the HYSPLIT (Hybrid Single Particle Lagrangian Integrated Trajectory; Stein et al.,
2015) model using the GDAS (Global Data Assimilation System) archived data sets.


**3 Results and discussion**

**3.1 Case study description**
During the campaigns there were a limited number of flights with the nucleation instrument
payload. Here we present a side by side comparison of two case studies, one from SPC (June 28,
2012) and the other from HTL (May 8, 2013). In addition the horizontal extent of NPF in SPC was
investigated by studying the research flight from June 30, 2012.

June 28, 2012 was a hot and sunny day in Po Valley. 24-h back trajectories arriving to SPC during
the morning revealed that the incoming air masses circulated from Central Europe and over the
Adriatic Sea before arriving to SPC from the southwest (Figure 2a). Figure 3 shows the time series
for some environmental parameters on the NPF event days from SPC and HTL. In SPC temperature
and RH showed a large diurnal variation; the temperature increased from 16 °C to 32 °C during the
morning while the RH decreased from 87% to 39%. The mean wind speed at 10 m height was 2.0 m





$s^{-1}$. These meteorological conditions and air mass histories are common during NPF event days in
Po Valley (Hamed et al., 2007; Sogacheva et al., 2007).

May 8, 2013 in HTL was a sunny and warm day with clear skies marked by broad diurnal variation
in temperature and RH. During the morning the temperature increased from 5 °C to 17 °C and the
RH decreased from 82% to 25%. The mean wind speed at 33.6 m height was 3.5 m $s^{-1}$. The air
masses originated from the North Atlantic Ocean arriving to HTL from the northwest via
Scandinavia and the Gulf of Bothnia (Figure 2b). Most NPF event days in HTL are clear sky days
with the arriving air masses spending most of their time in the northwest sector (Dada et al., 2017;
Nilsson et al., 2001; Sogacheva et al., 2008).

In SPC the solar radiation began to increase after 04:00 and the ML started to increase in height
around 06:00, at the same time the $SO_2$ concentration and $N_{>10}$ (number concentration of particles
larger than 10 nm) began to increase. This is likely explained by the entrainment of pollutants from
the RL and the onset of NPF. CS is higher during the night and decreases slightly during the day,
which is likely due to dilution related to ML growth.

At HTL after sunrise the $SO_2$ concentration and $N_{>10}$ decreased probably due to the dilution caused
by the growing ML coupled with the lack of pollution sources. While $SO_2$ concentration remained
low the whole day, $N_{>10}$ and CS began to increase later during the day because of the NPF event.
The average $SO_2$, $N_{>10}$ and CS in SPC were 0.57 ppb, 8102 $cm^{-3}$ and 0.0128 $s^{-1}$ respectively. While
in HTL the corresponding values were 0.02 ppb, 3293 $cm^{-3}$ and 0.0007 $s^{-1}$.

**3.2 Onset of NPF**
Figures 4a and 4b show the altitude of the Zeppelin as a function of time colored by the number
concentration of intermediate ions measured by the NAIS at SPC and HTL. The plots also show the
number concentration of intermediate ions measured on the ground as well as the ML height.

In SPC, the intermediate ion concentration began to increase on the ground at 5:48, which coincides
with the beginning of convective mixing and the breakup of the nocturnal surface layer. Similarly,
Kontkanen et al. (2016) observed that in Po Valley the onset of NPF coincided with the beginning
of boundary layer growth. Around this time the Zeppelin was profiling the layers above the ML.
"Pockets" of elevated intermediate ion concentration were present inside the RL (for example
around 700 m at 5:15). These pockets were not linked to the NPF event inside the ML. When the



Zeppelin later entered the ML at around 6:45, NPF was already taking place throughout the
developing ML and seemed to be confined to it.

In HTL, the number concentration of intermediate ions began to increase at around 6:47 on the
ground level. The ML at this point had grown to around 600 m above ground, which allowed us to
better resolve the onset of NPF vertically. In HTL no increase in intermediate ion concentration,
indicating no NPF, was observed above the ML on board the Zeppelin. Before 6:40 there was no
sign of NPF inside the growing ML. Between 6:40 and 7:00 the Zeppelin briefly measured in the
RL and re-entered the ML at 7:00. At this point the intermediate ion concentration was already
increasing on board similar to the ground level, indicating the onset of NPF.

Figure 5 shows the intermediate ion number concentration as a function of time from the Zeppelin
and the SMEAR II station. At the beginning of the NPF event, between 07:00-07:15, the Zeppelin
ascended from 300 m to 800 m. During the ascend the intermediate ion concentrations increased at
a similar rate and stayed at similar values on board the Zeppelin and at the ground level. The lack of
vertical gradient in the number concentration suggests that the aerosol particles were forming
homogeneously throughout the ML. However, intense turbulent mixing and strong updrafts moving
up at roughly the same rate as the Zeppelin might have also resulted in a homogeneous number
concentration, even if the aerosol particles were formed close to the surface.

Figures 4c and 4d show the Zeppelin's measurement profiles colored with the pseudo SA. In SPC,
the highest amount of pseudo SA appears to be in the residual layer above the growing morning ML
(also observed on June 27, 2012) after sunrise. This is in line with the observation that the $SO_2$
concentration increases at the surface when the ML starts to grow (Figure 3b), indicating that the
$SO_2$ was entrained from the RL. The entrainment of $SO_2$ from the residual layer is also supported by
previous observations (Kontkanen et al., 2016). The increased pseudo SA in the residual layer was
not associated with NPF in the residual layer.

In SPC the night time $SO_2$ concentration at the surface is low likely due to deposition (Kontkanen et
al., 2016). However ammonia concentration can be high (>30 µg m$^{-3}$) at the surface due to
agricultural activities and the concentration has been observed to peak during the night and early
morning (Sullivan et al., 2016).





Since in SPC the onset of NPF coincides with the beginning of ML growth, it is possible that the
entrainment of SA from the residual layer into the growing ML where ammonia, and likely also
amines from agricultural activities, are present can lead to stabilization of the SA clusters by the
ammonia and amines and subsequent NPF (e.g. Almeida et al., 2013; Kirkby et al., 2011).

In SPC the pseudo-SA layer closely corresponded to a layer of reduced condensation sink (CS). In
low CS regions more SA is in the gas phase and therefore detected by the APi-TOF (Figures 4e and
4f), which probably explains why the layer is there. In addition, the CS is also a sink for ions, which
means that the pseudo-SA is likely decreased even more than SA, assuming that the loss rate is
higher for ions than for SA molecules. By contrast, in HTL the amount of pseudo-SA is higher
inside the ML than above it. The pseudo-SA concentration increases on board throughout the
morning and peaks at roughly 9:00 and decreases afterwards.

In SPC pockets of intermediate ions and a layer of pseudo SA were observed in the RL, whereas at
HTL intermediate ion concentrations and pseudo SA remained low in the RL. This is likely related
to the relatively larger anthropogenic emissions in the Po Valley region compared to HTL. In
previous studies NPF has been observed inside the RL in Central Europe (Wehner et al., 2010) and
primary nanoparticles may be released into the RL from upwind pollution sources (Junkermann and
Hacker, 2018).

**3.2 Particle formation and growth rates**
Figure 6 shows the number size distributions measured by the NAIS on board the Zeppelin and on
the ground from SPC and HTL. The black dots are the mean mode diameters obtained by fitting a
log-normal distribution over the growing particle mode.

In SPC, the number size distributions measured on board and on the ground with the NAIS (Figures
6a and 6c) were similar when the Zeppelin was measuring inside the ML. When the Zeppelin
measured above the ML the number concentration decreased and the growing mode of freshly
formed particles was not observed. The pockets of intermediate ions in the RL did not grow to
larger sizes. This can be seen as sudden disappearances of the particles, for example at around 6:40,
7:15 and 8:00. The observations suggests that the NPF event was limited to the ML where it was
taking place homogeneously.





We calculated the formation and growth rates in SPC and HTL for particles and ions on board the
Zeppelin and on the ground. The results are summarized in Table 1. In SPC the onset of NPF
happened when the ML was still very shallow and the Zeppelin was not measuring significant
amount of time at this low altitude (this was a problem on other NPF event days from SPC as well),
consequently the beginning of the NPF event was not fully observed on board. Because of this we
were unable to reliably calculate the formation rates and the growth rate between 1-2 nm from the
Zeppelin data.

Kontkanen et al. (2016) obtained formation rates of 23.5 cm$^{-3}$ s$^{-1}$, 9.5 cm$^{-3}$ s$^{-1}$, 0.1 cm$^{-3}$ s$^{-1}$ and 0.08
cm$^{-3}$ s$^{-1}$ for 1.5 nm particles, 2 nm particles, 2 nm positive ions and 2 nm negative ions respectively
for the June 28, 2012 NPF event at the ground level. These values are in line with our values for the
same day reported in Table 1 ($J_3$ = 6.8 cm$^{-3}$, $J_3^-$ = 0.04 cm$^{-3}$, $J_3^+$ = 0.03 cm$^{-3}$). The higher formation
rates in SPC compared to HTL are characteristic of polluted environments (Kerminen et al., 2018).
The calculated GRs for the larger particle sizes as seen in Table 1 were similar on board the
Zeppelin (HTL: GR$_{7-20}$ = 2.4 nm/h, SPC: GR$_{7-20}$ = 3.0 nm/h) and on the ground (HTL: GR$_{7-20}$ = 2.1
nm/h, SPC: GR$_{7-20}$ = 2.8 nm/h).

On May 8, 2013 in HTL almost the whole NPF event was captured by the Zeppelin measuring
inside the ML. However, in contrast to SPC the number size distributions measured on board the
Zeppelin (Figure 6b) and on the ground (Figure 6d) show differences, particularly in the growing
nucleation mode particles. At different times on board the Zeppelin when it was measuring inside
the ML the particle number concentration in the growing mode momentarily increased up to eight
fold compared to the background number concentration, suggesting an enhancement in the particle
formation rate. On board the Zeppelin this can be seen as concentrated "vertical stripes" in the
number size distribution between 08:00-10:00. On the other hand at the ground station an increase
of concentration of freshly formed particles was observed between 7:30-8:00. This inhomogeneity
is further discussed in Section 3.3.

In the ground-based NAIS data a pool of sub-6 nm particles was present during the NPF event
while on board the Zeppelin no such pool was observed. This can be seen most clearly between
10:00-11:30 when the median particle number concentration between 2-4 nm on the ground was
1400 cm$^{-3}$ whereas on board the Zeppelin it was 570 cm$^{-3}$. Similarly Leino et al. (2019) observed
that the number concentration of sub-3 nm particles decreases as a function of altitude at HTL. This



may be linked to increased concentration of low-volatility vapors on the surface near the sources
compared to aloft.

Despite the differences in the ground-based and airborne number size distributions in HTL a
continuous, growing, nucleation mode was observed in the "background" both on the ground
(alongside the pool of sub-6 nm particles) and on board the Zeppelin during the NPF event. When
averaged over the total duration of the NPF event, the growth rates and formation rates on board the
Zeppelin and on the ground were similar on this day. This would indicate that the ground-based
measurements represent the NPF event in the whole ML quite well. However locally increased
number concentrations, indicating enhanced NPF, were observed inside the ML and if the
enhancement is not detected with the ground-based measurements we may underestimate the
intensity of NPF within the ML based on ground-based data alone.

**3.3 Vertical and horizontal distribution of the freshly formed particles**
Next we investigated how the freshly formed particles were distributed spatially in the BL. Figure 7
shows the particle number concentration between 3-10 nm measured by the NAIS and the ML
height from SPC as a function of time and altitude. The freshly formed particles were distributed
homogeneously throughout the growing ML but were not found in the RL. The 3-10 nm number
concentration inside the ML was ~20 000 cm$^{-3}$ while in the residual layer it was only ~200 cm$^{-3}$. The
pockets of increased intermediate ion concentration, indicating NPF in the nocturnal boundary layer
and residual layer (Figure 4a), were not observed in the 3-10 nm size range suggesting that the
particles did not grow to the 3-10 nm size range in any significant numbers.

At HTL the Zeppelin was measuring in the lower half of the developed ML, however the Cessna
profiled the entire depth of the ML all the way up to the lower parts of the free troposphere. Figure
8 shows the vertical profile of 3-10 nm particle number concentration between 07:00-10:00 UTC
calculated by subtracting the total SMPS number concentration from the UF-CPC number
concentration on board the Cessna. Also the water vapor concentration and temperature are shown.
A temperature inversion, a large negative gradient in water vapor concentration and in the particle
number concentration indicated that the top of the ML was present between 1300-1400 m.

On average the number concentration inside the ML remained roughly constant ($N_{3-10}$ ~1000 cm$^{-3}$)
as a function of altitude, however there was substantial variation (~200-3000 cm$^{-3}$). The strongest
variation came from a narrow sector roughly at the center of the measurement area, which is





discussed below. The NPF did not extend to the RL where the number concentrations were reduced
to below 100 cm$^{-3}$.

However at 2000 m a layer of sub-10 nm particles was observed. The 3-10 nm number
concentration increased from less than 100 cm$^{-3}$ to ~400 cm$^{-3}$. Lampilahti et al. (2020a) showed
evidence that NPF frequently takes place in the interface between the residual layer and the free
troposphere, disconnected from the ML. Precursor gases may be transported to these altitudes and
the mixing over the interface layer could initiate nucleation.

Figure 9a shows the particle number concentration between 3-10 nm on board the Zeppelin and the
airplane as a function of longitude and latitude from HTL on May 8, 2013. The particle number
concentration was elevated right over HTL in a narrow sector perpendicular to the mean wind
direction. Vertically the sector extended throughout the depth of the ML. The number concentration
in the sector increased 2-8 fold compared to the surrounding background number concentration. The
mean wind speed in the ML was about 4 m/s and the particle sector was observed throughout the
whole measurement flight, for at least 2.5 hours. This suggests that the particle sector was probably
at least 35 km long along the mean wind direction.

The concentrated vertical stripes over the growing nucleation mode in Figure 6b were caused by the
Zeppelin periodically flying through the particle sector. The sector slowly moved perpendicular to
the mean wind towards northeast and when passing over HTL it was seen as the plume of particles
in Figure 6d between 07:30-08:00. The particles in the sector grew at approximately the same rate
with the background NPF event particles, which also suggests that the particles were formed
simultaneously inside the long and narrow sector. Lampilahti et al. (2020b) showed that these types
of NPF events, or local enhancements of regional NPF events, are common in HTL and that they
are linked to roll vortices, which are a specific mode of organized convection in the BL.

On June 28, 2012 in SPC the Zeppelin flew the measurement profiles over a small area and
therefore it was difficult to infer the horizontal extent of the NPF event. However, on June 30, 2012
the Zeppelin measured over a larger area in order to find the edges of the airmass where the NPF
event was taking place. The flight on June 30, 2012 lasted from 05:00 to 10:00 UTC. Figure 9b
shows that the NPF event was observed to occur in the sector of the Valley comprised between
Ozzano (just north of the Apennine foothills) and the city of Ferrara (just south of the Po river). The



area in between experienced westerly winds, from the inner Po Valley toward the Adriatic sea,
which is a common feature of the Po Valley wind breeze system in the early morning.

Farther north of the Po river, an easterly breeze was developing and no NPF was observed (off the
map in Figure 9b, see Figure 10). Nocturnal north-easterly breezes are often observed over the
Three Venezie Plain as a result of a low-level jet (Camuffo et al., 1979). The variability in local
wind fields may generate chemical gradients in the atmospheric surface layer within the Po Valley,
hence segregating air masses which can be active or inactive with respect to NPF, in complete
absence of orographic forcings (i.e. over a completely flat terrain). Probably the air masses with an
easterly component reaching the Zeppelin from the Venetian plain picked up pollution (e.g. CO,
$NO_x$) from urban sources, but we can also speculate that for example ammonia and amines were
much lower than in the westerly air masses flowing south of the Po river, which had crossed the
areas between Emilia and Lombardy where most agricultural activities take place (see Figure 10). A
chemical transport model run predicting $NH_3$ concentrations with adequate resolution, and using
them as a tracer for the actual precursors for NPF, might clarify this point. However modeling
atmospheric transport at this scale in an environment like Po Valley can have substantial
uncertainties (Vogel and Elbern, 2021).


**4 Conclusions**

Flight measurements are essential to evaluate the representativeness of the ground-based in-situ
measurements. In many cases it may be impossible to tell from only ground-based data what drives
the observed NPF, especially when the effect of BL dynamics is important. Atmospheric models
require field observations for validation and constraints. Airborne measurements such as the ones
reported here provide valuable data for this purpose.

We compared two different environments where NPF occurs frequently: a suburban area in Po
Valley, Italy, and a boreal forest in Hyytiälä, Finland. We aimed to answer in which part of the BL
the onset of NPF and the growth of the freshly formed particles takes place and studied the vertical
and horizontal extent of NPF.

To detect directly the very first steps of NPF in the BL, we used airborne Zeppelin and airplane
measurements, supported by ground-based in-situ measurements. The Zeppelin measurements



allowed us to study the vertical extent of NPF in the BL. The high time resolution and low cut-off
size of the instruments on board allowed us to observe the starting time, location and altitude of an
NPF event.

Within the limits of the Zeppelin's vertical profiling speed (~ 0.5 m/s ascend) and the time
resolution of the NAIS, we observed that the onset of NPF happened simultaneously inside the ML.
However particles formed close to the surface could probably still be mixed by strong updrafts fast
enough so that the number concentrations measured on board the Zeppelin appear homogeneous.
The newly formed particles were observed to grow to larger sizes at the same rate within the ML.
However, in HTL we observed local enhancements in NPF that were induced by roll vortices in the
BL.

In addition a separate layer of sub-10 nm particles was observed above the ML in HTL. Lampilahti
et al. (2020b) showed that such layers in HTL are likely the result of NPF in the topmost part of the
RL. Furthermore it was estimated that around 42% of the NPF events observed in HTL at the
surface are entrained from such elevated layers. In SPC we observed how NPF could be happening
in one air mass but be completely absent in an adjacent air mass with a different origin.

We presented three case studies (two from Italy and one from Finland). The conditions on our case
study days represent the typical conditions in these locations when NPF events usually occur. That
is to say, a sunny day with the air masses originating from a certain area during a period of the year
(May in HTL and June in SPC) when NPF is common. Nevertheless it is not certain that our case
studies represent all NPF event days. NPF events also occur under different kinds of conditions. The
growing nucleation mode particles originating from NPF do not always grow smoothly and
continuously in the measured size distribution like in our cases, but may have large variation and
discontinuities, which may reflect the vertical and horizontal variability in NPF.

**Acknowledgements**
This research was supported by the European Commission under the Framework Programme 7
(FP7-ENV-2010-265148). The support by the Academy of Finland Centre of Excellence program
(project no. 272041 and 1118615), the ERC-Advanced "ATMNUCLE" (grant no. 227463), the
Eurostars Programme (contract no. E!6911), and the Finnish Cultural Foundation is also gratefully
acknowledged. The Zeppelin is accompanied by an international team of scientists and technicians.
They are all warmly acknowledged.




**Data availability.** Ground-based meteorological data, radiation, gas and particle size distribution

data from HTL is available from https://smear.avaa.csc.fi/ (last access: Apr 1, 2021). The Cessna

dataset is available from https://doi.org/10.5281/zenodo.3688471 (last access: Oct 23, 2020). The

rest of the data used was gathered into another dataset: https://doi.org/10.5281/zenodo.4660145.

**Author contributions.** HM, TN, SM, ME, IP, SS, JKa, EJ, TYJ, RK, KLeh, SD, AM, RT, DW, FR,

TP, TM and MK coordinated the Zeppelin campaign. RV carried out the Cessna measurements. JLa,

TN, HM, JKo, KLei and VMK analyzed and interpreted the data. JL and HM prepared the

manuscript, with contributions from all coauthors.

**The authors declare that they have no conflict of interest.**

551





552

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

554



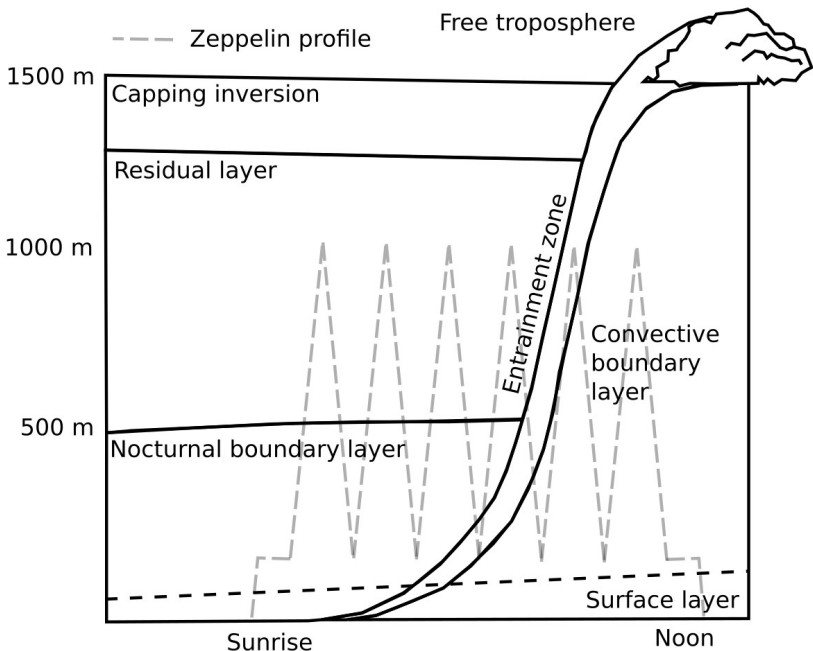

Figure 1: A schematic diagram of different atmospheric layers in the lower troposphere and their development during the morning hours. A generic Zeppelin measurement profile (dashed gray line) is displayed on top. The figure is adapted from Stull (1988).



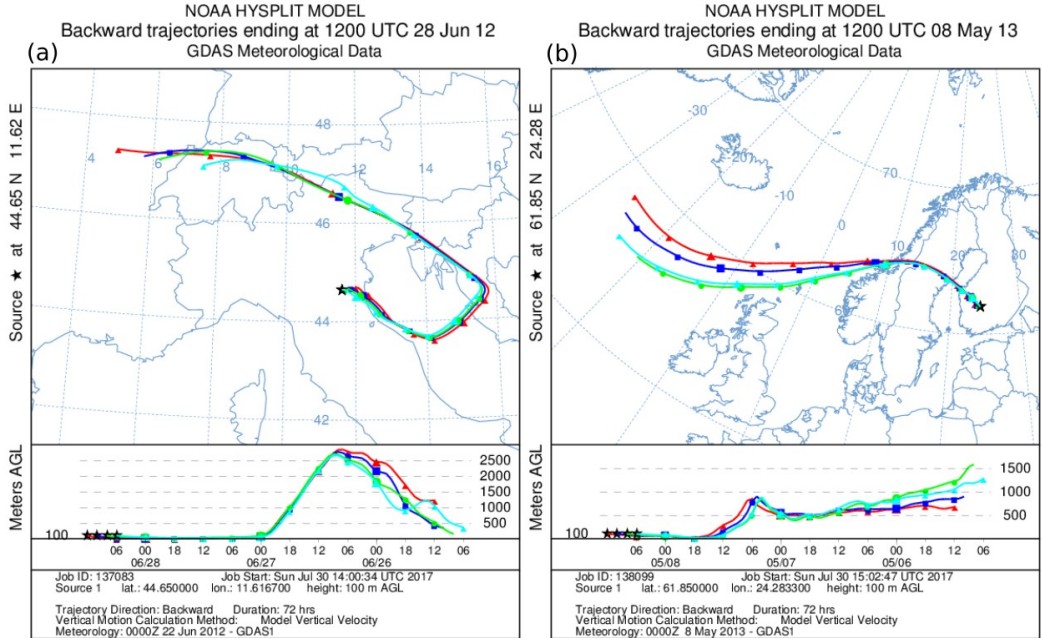

559     Figure 2: Airmass backward trajectories to (a) SPC during the morning of June 28, 2012 and (b)
560     HTL during the morning of May 8, 2013.





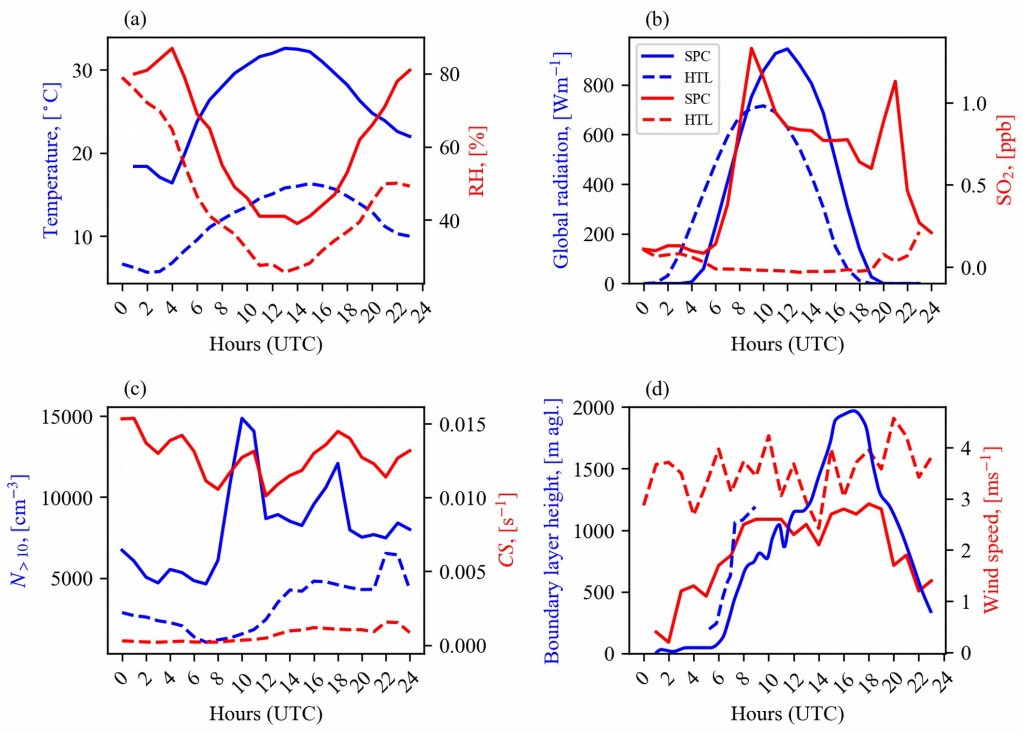

Figure 3: Diurnal variation in (a) temperature, relative humidity, (b) global radiation, SO2 concentration, (c) >10 nm particle number concentration, condensation sink (CS) and (d) mixed layer height in SPC on June 28, 2012 and in HTL on May 8, 2013.



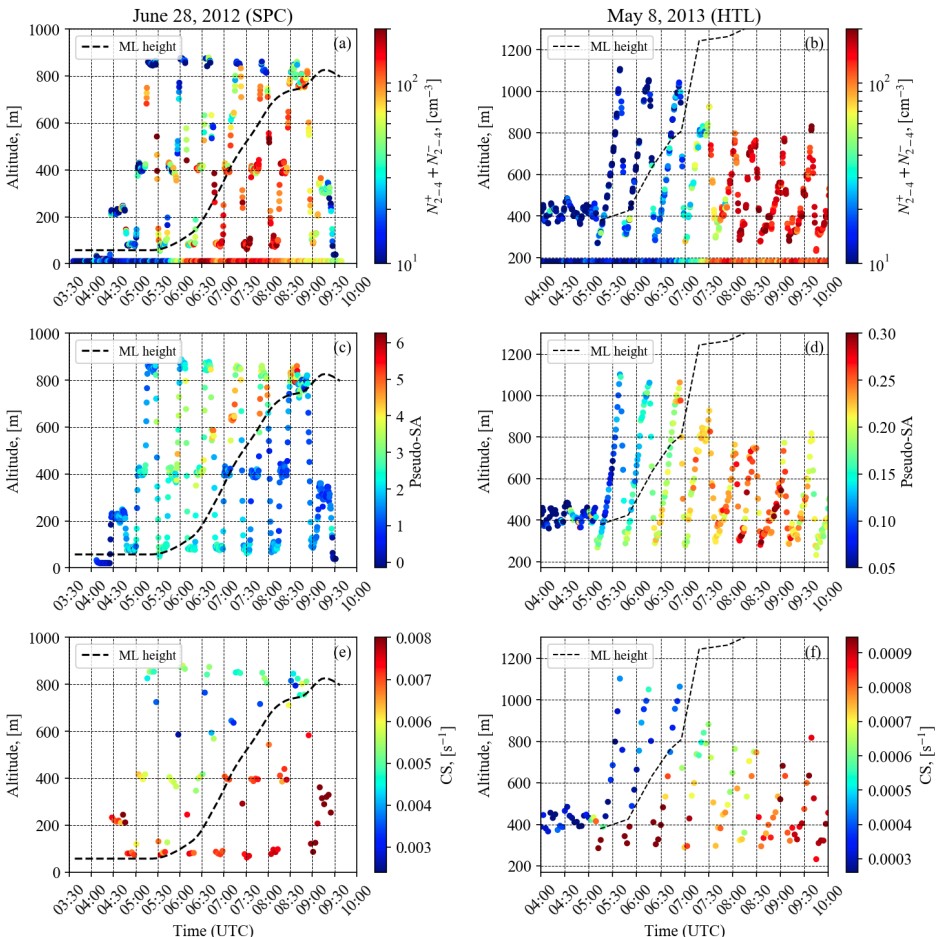

Figure 4: Time-evolution of selected variables as a function of height in SPC and HTL. Panels (a) and (b) show the intermediate ion number concentration from SPC and HTL. Ground-based measurements as well as measurements from the Zeppelin are shown. Panels (c) and (d) show the pseudo-SA from SPC and HTL. Panels (e) and (f) show the CS. Height of the mixed layer is shown in all panels.

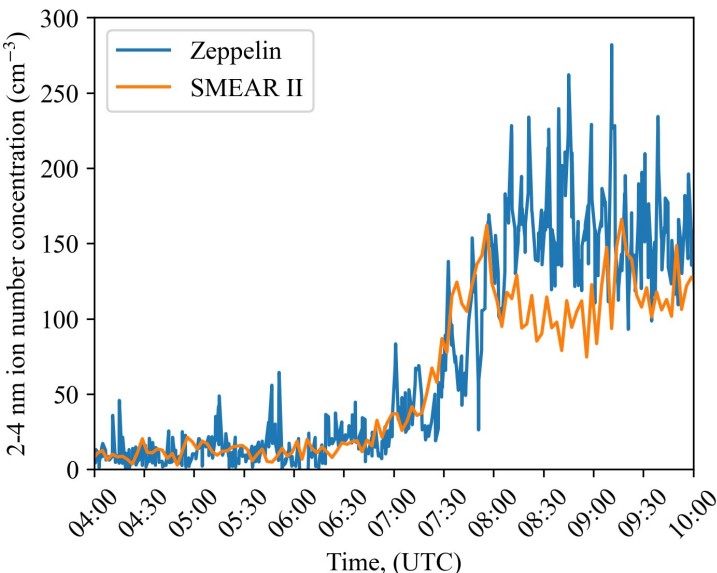

Figure 5: Time series of intermediate (2-4 nm) ion number concentration on board the Zeppelin and the SMEAR II station in HTL on May 8, 2013.







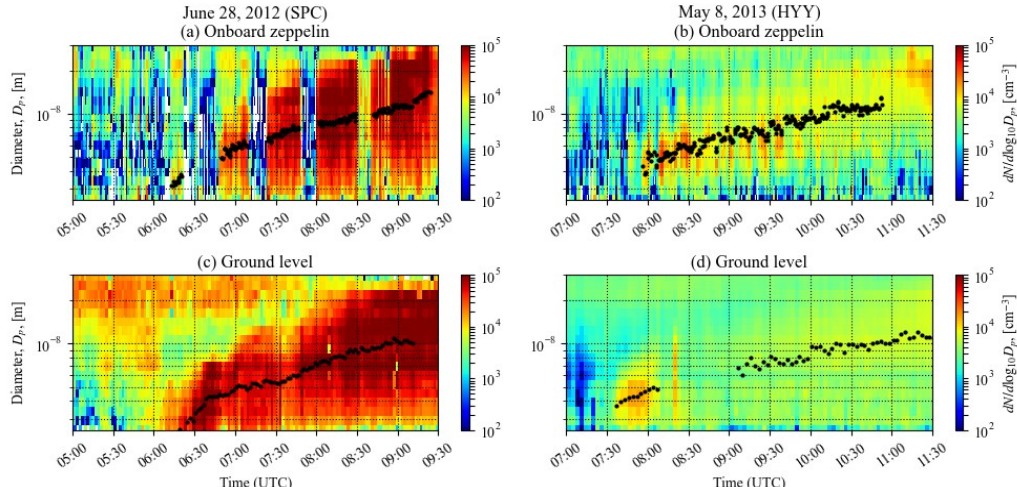

Figure 6. Time evolution of particle number size distributions measured by the NAIS (positive
polarity) on board the Zeppelin (a, b) and at the ground level (c, d) in HTL and in SPC on the two
case study days. The black dots are the mean mode diameters found by fitting a log-normal
distribution over the growing mode.




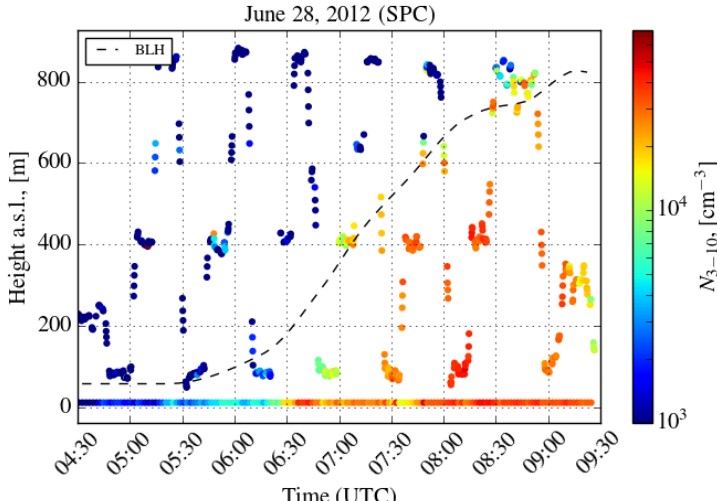

Figure 7. The particle number concentration in the 3-10 nm size range from SPC on board the
Zeppelin and on the ground level on June 28, 2012. BLH refers to the boundary layer height
determined from ceilometer.



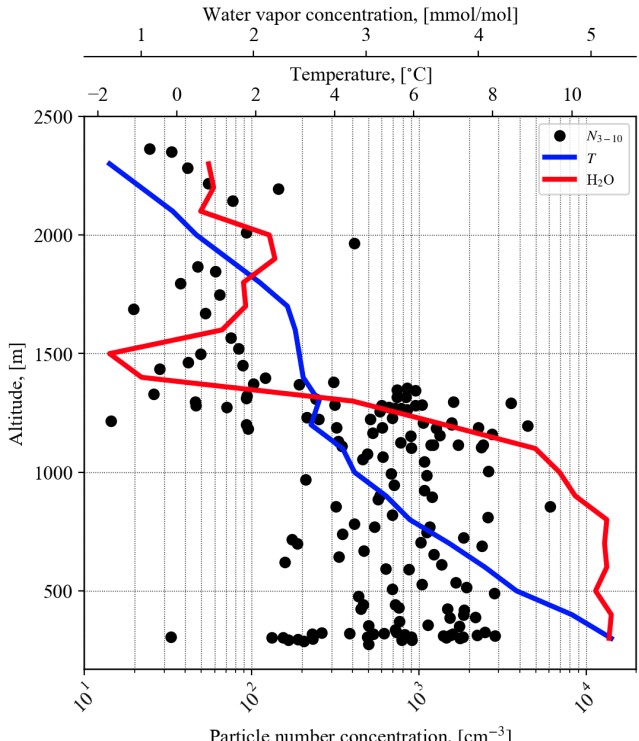

Figure 8: Vertical profile of 3-10 nm particle number concentration (black dots), temperature (blue line) and water vapor concentration (red line) measured on board the Cessna between 07:00-10:00 on May 8, 2013 in HTL.



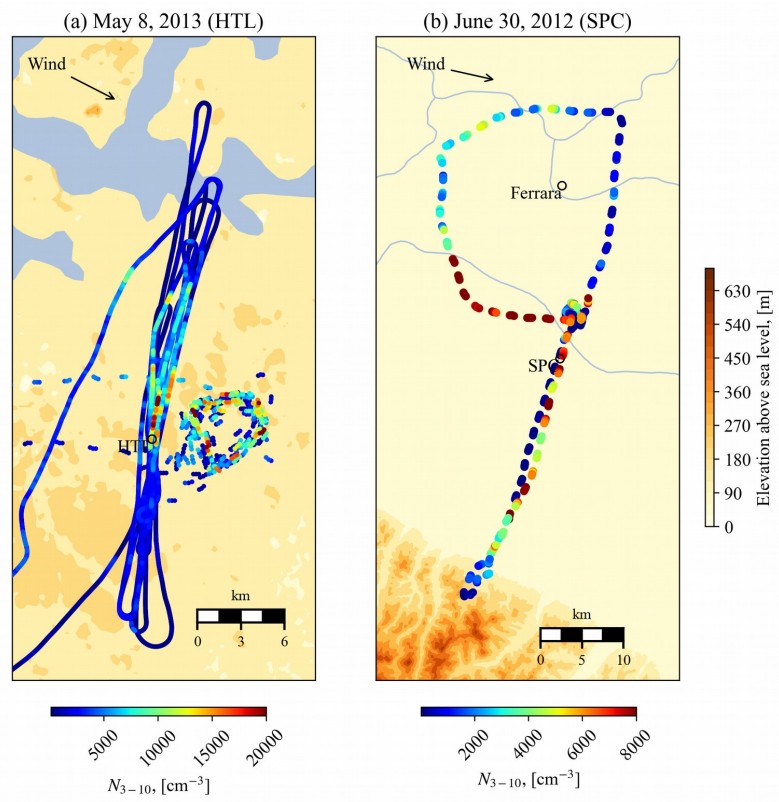

Figure 9: (a) the flight tracks of the Zeppelin (circular track) and the airplane (track with back an
forth segments) colored by 3-10 nm particle number concentration from HTL on May 8, 2013. (b)
the flight track of the Zeppelin colored by 3-10 nm particle number concentration from SPC on June
30, 2012. The Zeppelin flight track has gaps because the NAIS was measuring in the ion mode
during that time.



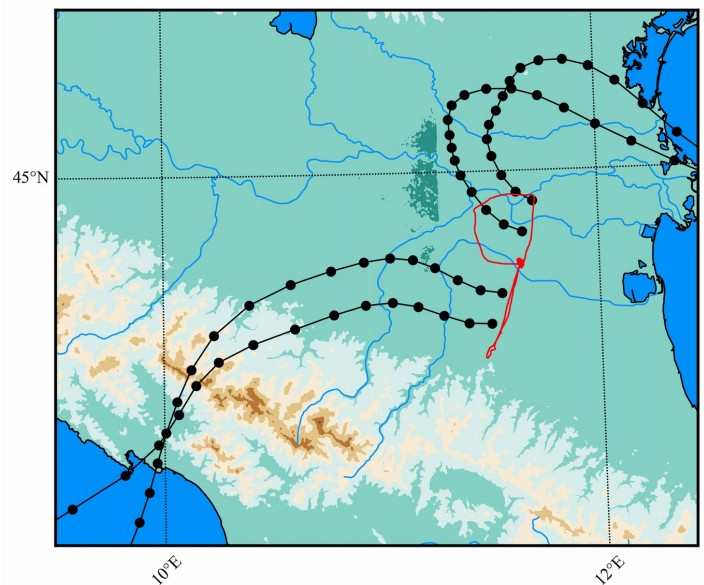

Figure 10: Airmass back trajectories (black dotted lines) arriving to the Zeppelin's measurement
area over north Italy on June 30, 2012. The separation between the dots along the trajectories is one
hour. The red line is the Zeppelin's flight track.





Table 1. Calculated particle formation and growth rates. + and – superscripts refer to positive and
negative ions respectively. The Zeppelin missed the beginning of the NPF event in SPC and because
of that some values are missing.

|  | HTL (May 8, 2013) | | SPC (June 28, 2012) | |
|---|---|---|---|---|
|  | Zeppelin | Ground | Zeppelin | Ground |
| $J_{1.5}$, [cm$^{-3}$ s$^{-1}$] | 1.5 | 0.9 | - | - |
| $J_3$, [cm$^{-3}$ s$^{-1}$] | 0.2 | 0.3 | - | 6.8 |
| $J_3^-$, [cm$^{-3}$ s$^{-1}$] | 0.04 | 0.04 | - | 0.04 |
| $J_3^+$, [cm$^{-3}$ s$^{-1}$] | 0.04 | 0.04 | - | 0.03 |
| $GR_{1-2}$, [nm h$^{-1}$] | 0.8 | 0.7 | - | 0.5 |
| $GR_{2-3}$, [nm h$^{-1}$] | 1.4 | 1.5 | 1.8 | 1.5 |
| $GR_{3-7}$, [nm h$^{-1}$] | 1.7 | 1.6 | 2.9 | 2.0 |
| $GR_{7-20}$, [nm h$^{-1}$] | 2.4 | 2.1 | 3.0 | 2.8 |
