# Peer review of "Zeppelin-led study on the onset of new particle formation in the planetary boundary layer"

_Atmospheric Chemistry and Physics, 2021_

## Author Response (AR1)

This paper presents measurements of new particle formation above ground using a Zeppelin in two different environments in Europe, in northern Italy and in Finland. This is an excellent set of measurements demonstrating that these events take place uniformly in the boundary layer in both locations. The paper is well written and should be published after the authors address the following issues.

Major issues

(1) The major weakness of the paper is that it does not even mention the results of the other Zeppelin flights and the ground measurements during the two campaigns. I do appreciate the in depth analysis of the case studies, but it is not clear if there represent what happens most of the time in these two areas or if they are very special days. I think a section summarizing the results of all flights and their similarities (or differences) with the case studies discussed is needed.

ANSWER: The detailed measurements with NAIS and APITOF were only available from the nucleation layout flights. These included 5 flights in Italy and 6 flights in Finland. The NPF event was fully captured on only some of those days, which leaves us with the case studies.

Even though the time of the year and meteorology represent a situation when NPF usually happens in Po Valley and Hyytiälä, we acknowledge that the case studies may not represent the typical case of NPF. We will try to emphasize this more in the introduction and conclusions, stating that the results are from case studies.

Studying the average profile from the roughly 30 flights in Po Valley and Hyytiälä using the SMPS data and comparing it to other measurements is probably best done in a separate manuscript.

(2) Despite the presence of relatively high levels of sulfuric acid in the residual layer above the Po Valley there was no NPF there (Figure 4c). This is an interesting observation that deserves some discussion and discussion. I understand that the Zeppelin was not measuring the concentrations of gas-phase pollutant during this flight but my understanding is that the authors have some measurements during other flights in the campaign. What was different in the RL? They suggest that may be there was not enough ammonia there. However, the presence or lack of VOCs is probably worth some discussion using the observations of VOCs in that region during other flights in the campaign.

ANSWER: We added the following piece of discussion about VOCs:

"In addition oxidized VOCs are important for aerosol particle growth (Ehn et al., 2014). VOCs were measured on board the Zeppelin in Po Valley in 2012 and the results showed higher VOC concentrations close to ground Jäger (2014). This may at least partly explain why we measured increased concentrations of intermediate ions in the RL but they did not grow to larger sizes in any significant quantities."

(3) I was surprised by the measured spatial extent of NPF in Hyytiala. According to the measurements it is taking place in a relatively narrow area of 30-40 km around the station and not over scales of 100s of kilometers as it has been sometimes assumed. However, there is little discussion of what is happening in this relatively narrow corridor that leads to NPF and what is

missing outside it and NPF is not happening. To be more provocative are all of these NPF observations over the years in Hyytiala referring to something that is quite limited in space and covers only a small fraction of the boreal forest?

ANSWER: We studied this phenomenon further in a separate paper that was published in 2020 and found that these narrow zones of NPF seem to be related to locally enhanced NPF caused by organized convection in the BL, more specifically roll vortices (Lampilahti et al 2020).

(4) There is little discussion of the measurements of the composition of the smallest particles during these flights.

ANSWER: The composition of the particles in the sub-20 nm range could not be determined with the instruments on board. With the APiTOF we were able to detect [HSO4-] ions and used it as an estimate for sulfuric acid in the gas phase as this only required one or two distinct peaks that were relatively easy to spot. However due to low signal and changes in pressure, other interesting compounds like organic molecules could not be reliably detected and this data was not included in the manuscript.

Minor points

(5) I had some difficulty with Figure 3b (SO2 in Hyytiala) and Figure 3c (CS in Hyytiala) until I realized that the y-axis includes negative concentrations. I strongly suggest starting these axes from zero. Also does the N axis in Figure 3c start from zero or from another value?

ANSWER: Changed the axis to start from zero

(6) The legend of Figure 3 should mention that these are ground measurements.

ANSWER: We added this to the caption.

References

Ehn, M., Thornton, J. A., Kleist, E., Sipilä, M., Junninen, H., Pullinen, I., Springer, M., Rubach, F., Tillmann, R., Lee, B., Lopez-Hilfiker, F., Andres, S., Acir, I.-H., Rissanen, M., Jokinen, T., Schobesberger, S., Kangasluoma, J., Kontkanen, J., Nieminen, T., Kurtén, T., Nielsen, L. B., Jørgensen, S., Kjaergaard, H. G., Canagaratna, M., Maso, M. D., Berndt, T., Petäjä, T., Wahner, A., Kerminen, V.-M., Kulmala, M., Worsnop, D. R., Wildt, J., and Mentel, T. F.: A large source of low-volatility secondary organic aerosol, Nature, 506, 476–479, https://doi.org/10.1038/nature13032, 2014.

Jäger, J.: Airborne VOC measurements on board the Zeppelin NT during the PEGASOS campaigns in 2012 deploying the improved Fast-GC-MSD System, Forschungszentrum Jülich GmbH, 2014.

Lampilahti, J., Manninen, H. E., Leino, K., Väänänen, R., Manninen, A., Buenrostro Mazon, S., Nieminen, T., Leskinen, M., Enroth, J., Bister, M., Zilitinkevich, S., Kangasluoma, J., Järvinen, H., Kerminen, V.-M., Petäjä, T., and Kulmala, M.: Roll vortices induce new particle formation bursts in the planetary boundary layer, 20, 11841–11854, https://doi.org/10.5194/acp-20-11841-2020, 2020.

This paper describes new particle formation event observed in a rural (SMEAR 2 station) and urban (Po Valley) environment using instruments installed on the ground and in a Zeppelin. As claimed by the authors, "the main goal of the manuscript was to quantify the magnitude of regional to global feedbacks between atmospheric chemistry and physics, and thus quantify their impact on the changing climate". However, I don't think this was achieved within this manuscript. Moreover, it's not clear to me why the authors choose to compare event at both sites. Of course, both environments lead to different situations favorable for NPF. However, the authors are comparing one study case for each environment picked out from short term campaigns. Why these two cases are representative of NPF events in both environment?

ANSWER: The paragraph in the introduction that says

"Here we present NPF measurements on board a Zeppelin airship carried out during the EU supported PEGASOS (Pan-European Gas-AeroSOls Climate Interaction Study) project. The main goal of the project was to quantify the magnitude of regional to global feedbacks between atmospheric chemistry and physics, and thus quantify their impact on the changing climate. The Zeppelin flights were used to observe radicals, trace gases, and aerosol particles inside the lower troposphere over Europe in several locations during 2012-2013"

is referring to the PEGASOS project goals. The main goals of the current manuscrtipt are:

"Here we combine comprehensive ground-based and airborne measurements from the Zeppelin to compare two NPF cases from Po Valley to one case from Hyytiälä. The Zeppelin allowed us to repeatedly profile the lowest 1 km of the atmosphere providing a full picture of what is happening in the BL during the onset of NPF. We will show in which part or parts of the BL the onset of NPF and the subsequent particle growth occurred at the two measurement sites as well as determine formation and growth rates for the aerosol particles."

We removed the sentence "The main goal of the project was to quantify the magnitude of regional to global feedbacks between atmospheric chemistry and physics, and thus quantify their impact on the changing climate." from the first paragraph so that there would be no confusion.

Detailed measurements during NPF events using the nucleation payload instruments were only obtained on these couple of days. We do not argue that the NPF events we measured represent how the average NPF events occur in the BL, we only point out that the timing and meteorological conditions were typical of NPF events at these locations. Despite this limitation we believe this manuscript gives valuable insight to our understanding on how BL processes can affect NPF at the studied locations.

Major comments

I regret that the authors cite papers mostly from their group. There are other high quality papers out there working on NPF events analysing the vertical extension of NPF event, looking at the link with turbulences…

Figure 5 : I'm not sure I agree with the author's conclusions on that figure. On the SPC side, there are Ultra Fine (UF) particles in the residual layer during the early morning even before the sun rise

(4:45). Moreover, during the whole morning the UF particules can be seen at the top of the ML within the ML and in the TL. The NPF onset at the ground is clearly between 5:30 and 6:00. At 5:00, measurements from the zeppelin are showing high concentration of UFP below 200m. It looks like this specific event may be caused by turbulence mostly. On the other side (HTL), clearly the NPF event started at the ground at 7:30. There are no signs of UFP at the ML top but some UFP lies at 2km according to CESSNA measurements. The authors claimed (L. 507/L. 433), based on previous publications, that event occurring at 2km could be linked to turbulence occurring between residual layer and free troposphere and that both events (2km and ground) could be linked but I don't understand how these could be linked… Please do tell ! Moreover, there are no evidence that the interface RL/FT is located at 2km, nor that both events are linked…

ANSWER: There seems to be some confusion with the figure numbers, I believe this is referring to Fig 4.

We agree that in the SPC case turbulent mixing plays a crucial role in the onset of NPF. In the manuscript we suggest that when turbulent mixing starts and the ML begins to grow sulfuric acid, which appears to be concentrated in the RL, is mixed to the surface from the RL. At the surface ammonia and other precursor gases like amines and organics are present and in the resulting mixture NPF starts.

In HTL case we argue that the sub-10 nm particles at 2 km observed from the CESSNA are not linked to the NPF event inside the ML on that day, instead they are probably two separate NPF events. The atmospheric layers above the ML are not clearly visible during the Cessna profile, however observations of such particle layers have been linked to the RL-FT interface over Hyytiälä and therefore we provide it as a possible explanation.

There are no date in Figure 5 label and I think this is missing…

ANSWER: We added the dates in the caption of the Figure (assuming again it is Figure 4)

L 388 : « In the ground-based NAIS data a pool of sub-6 nm particles was present during the NPF event… This can be seen most clearly between 10:00-11:30 .. ”

I believe you were referring to Figure 7b and d at HTL (not HYY as written on the figure).  I don't clearly see it from 10:00-11:30 but I do see it during the whole day, right? Again, this rises question about how the events were created. So in SPC, the particle mode is really large always including 3nm particles during the full day suggesting that the event is all over the ML and therefore is not comparable to the HTL event that is observed at the ground and newly formed particles grow and may be then transported into the ML.

ANSWER: It seems that in HTL the formation of the smallest particles might be increased close to the forest canopy, which makes sense since the boreal forest is a source of the organic precursor vapors. Leino et al. (2019) observed increased sub-3 nm particle concentrations towards the surface in HTL even on days when no NPF event occurred. This particle concentration gradient was less pronounced during NPF event days. In HTL the particles could still form throughout the ML, but the formation rate could be slightly higher close to canopy. We changed HYY to HTL in the figure.

L 445 – 452 "The concentrated vertical stripes over the growing nucleation mode in Figure 6b were caused by the ….Are linked to roll vortices, which are a specific mode of organized convection in the BL. " Could you please take some time to prove it ? As you are comparing two events in different environment I think that you should carefully address how those events appear …

ANSWER: The location and movement of the particle zone matched that of roll vortices over the measurement area (a boundary layer deep longitudinal zone, less than few km wide, that moved over the measurement site perpendicular to mean wind). This case was analyzed in the paper cited in the manuscript (Lampilahti et al., 2020). In that paper specifically look at Figure 11. More analysis related the Figure 11 was presented in a response to a reviewer (https://doi.org/10.5194/acp-2019-1013-AC1: see figures 3, 4 and 5).

Conclusions :

"We compared two different environments where NPF occurs frequently: a suburban area in Po Valley, Italy, and a boreal forest in Hyytiälä, Finland. We aimed to answer in which part of the BL the onset of NPF and the growth of the freshly formed particles takes place and studied the vertical and horizontal extent of NPF. "

Again why choosing two different environments and only two study cases to answer that question ? I would think that more statistical information would be needed …

ANSWER: We ackowledge that these observation may not represent the typical NPF event at the sites. To make it more clear we rephrased the paragraph to read:

"We compared case studies from two different environments where NPF occurs frequently: a suburban area in Po Valley, Italy, and a boreal forest in Hyytiälä, Finland. We aimed to answer in which part of the BL the onset of NPF and the growth of the freshly formed particles took place and studied the vertical and horizontal extent of NPF."

Minor remarks

P4 L 104 : rephrase : 'Compare from nucleation '

ANSWER: Changed the whole sentence to: "The two ground-based measurement sites that were studied here were San Pietro Capofiume in Po Valley, Italy and Hyytiälä in Southern Finland."

P9 L 272 : "ML started to increase in height" could you please highlight where this is coming from ? Lidar, ceilometer measurement of in flight measurements ?

ANSWER: Yes, changed to: "according to the ceilometer measurements the ML started to increase in height"

P9 L 300 : "no NPF, was observed above the ML" : remove the comma

ANSWER: Done

Figure 6 is hard to read. The Zeppelin measurements were performed at different altitude and it does not appear. Could you please either remove it either include the altitude on that figure ?

ANSWER: The altitude is now included in the figure.

Figure 7 : always add the altitude to any on-board measurements

ANSWER: The altitude is now included in the figure.

Figure 7d : from 8am to 9am there is no black dot. So the GMD is over 30nm ? so there is a clear interruption of this event ! Could you comment on that ? What does that change for your study or for the event in general ??

ANSWER: At the SMEAR II station between 7:30-08:00 we observed the localized NPF event that was linked to the organized convection and moved over the station with the airmass. The black points after 09:00 were linked to the regional NPF event that was taking place at the same time, the mode fitting method could not determine a clear mean mode diameter before 09:00 for the regional NPF event.

References

Leino, K., Lampilahti, J., Poutanen, P., Väänänen, R., Manninen, A., Buenrostro Mazon, S., Dada, L., Franck, A., Wimmer, D., Aalto, P. P., Ahonen, L. R., Enroth, J., Kangasluoma, J., Keronen, P., Korhonen, F., Laakso, H., Matilainen, T., Siivola, E., Manninen, H. E., Lehtipalo, K., Kerminen, V.-M., Petäjä, T., and Kulmala, M.: Vertical profiles of sub-3 nm particles over the boreal forest, 19, 4127–4138, https://doi.org/10.5194/acp-19-4127-2019, 2019.

Lampilahti, J., Manninen, H. E., Leino, K., Väänänen, R., Manninen, A., Buenrostro Mazon, S., Nieminen, T., Leskinen, M., Enroth, J., Bister, M., Zilitinkevich, S., Kangasluoma, J., Järvinen, H., Kerminen, V.-M., Petäjä, T., and Kulmala, M.: Roll vortices induce new particle formation bursts in the planetary boundary layer, 20, 11841–11854, https://doi.org/10.5194/acp-20-11841-2020, 2020.

Comment on the manuscript:

Zeppelin-led study on the onset of new particle formation in the planetary boundary layer by J. Lampilahti et al.

The manuscript describes airborne data from the Zeppelin aerosol measurements during the Pegosos Campaigns in San Pietro Capofiume in 2012 and in Hyytiälä in 2013. The results are highly interesting and could be a useful data set to trace back the appearance and origin of nucleation mode particles in the atmosphere or, to characterize air masses that contain precursor material, as proposed in the abstract. In both cases, in SPC and in HTL the data on the scale of the spatial distribution suggest a significant contribution of horizontal transport on top of the diurnal cycle of vertical convection. An interesting new result is the signature of sulphuric acid found above the MBL before convection mixed a larger MBL volume. Getting into more detail a chemistry transport model would be useful, not included in the current study.

A further well notified result is, that despite transport over major agricultural areas in the center of the eastern Po-Valley, west of Ferrara enhanced nucleation aerosol was not observed. This is a bit surprising after to the results of Kontkanen et al, 2016, who found NPF events each day with only one exception, a paper suggesting that NPF is more a general feature in the Po-Valley.

However, focusing on the current manuscript, the title promises new results on the

'onset of new particle formation in the planetary boundary layer usig an airborne platform (Zeppelin)'

For such an investigation a precise time and location of the data points would be necessary. Unfortunately these data are not presented and on top there are several obvious timing problems clearly visible in the figures:

Fig. 2, Trajectories were calculated for 12:00 UTC (see HYSLIT info in the plots, not for the morning as claimed in the figure caption). Already 2 hours difference might be critical for the wind direction and trajectory. Also, for a process occurring in a diurnal cycle with short lived compounds a trajectory for 72 h does not make a lot of sense.

ANSWER: The information in the HYSPLIT figures is a bit confusing, the different colored trajectories are for different hours of the morning. We updated the figure to address this.

Fig.3, Time axis in the diurnal plots, is claimed to be 'UTC'. The geographical location of San Pietro Capofiume is 44°39'N,11°37'E. This should result in a peak solar radiation about one hour (1h for 15o) earlier than 12:00 UTC (see also Kontkanen et al 2016, where the same radiation data are presented with the correct timestamp, UTC+1)

ANSWER: This makes sense, even though the data readme says UTC the data is probably in UTC+1 (local time). We shifted the time by 1 hour backwards in the diurnal variation plots where it was needed (meteorology, radiation and SO2 concentration).

Fig. 4, timing problem in the MBL development. Despite less incoming radiation the MBL growth in Hyytiälä is faster than in SPC. A more realistic temporal evolution of the MBL using HYSPLIT data results in a MBL of +1200 m at ~ 09:15 UTC instead of 07:15 UTC. This finally leads to a different interpretation of the HTL data.

ANSWER: The figure below shows the Cessna climb between 7:30-7:50 UTC. The top of the BL indicated by the temperature inversion and stable layer appears to be at 1200 m asl, already before 9:15 UTC. The BL height at 7:15 was determined from a Cessna profile where the airplane climbed only to about 1400 m asl and the top of the BL was not as clear, so we left this first BLH data point from the Cessna out. This does not lead to different interpretation of the results though.

[Figure]

Fig, 6, a) SPC, position data for June 28 are missing, compared to Fig. 9, horizontal position is important. b) HTL Data in Fig. 9 do not support the conclusions in the text. Low wind situation with a good chance to get into self-contamination problems in circle flights. The time spent for half a circle (20 m/sec and ~4 km diameter) is close to air mass transport time across circle. Timing requirements for such a local case study are even more stringent than just using the correct time zone. Even the internal timing of the NAIS or SMPS scanning loop becomes important (Manninen et al, 2016). Is the observation of 3-10 nm particles in agreement with the growth rate and diurnal pattern of > 10 nm particles at HTL in Fig. 3?

ANSWER: Unlike in HTL no clear horizontal variation was observed in SPC on June 28, 2012 on board the Zeppelin. The Zeppelin was measuring inside a small area less than few km in diameter in order to capture mostly vertical variation. Adding a horizontal Zeppelin track for June 28, 2012 would not bring much new information which is why we do not include it.

We do not think there was a problem with self-contamination on May 8, 2013 since the zone of sub-10 nm particles was observed on board the Zeppelin, Cessna and at the SMEAR II station, separated more than couple of km from each other. Also the particle zone was not more concentrated in the downwind section of the Zeppelin's flight track, instead it appeared to move slowly approximately

from soutwest to northeast (see Figure 3 in https://doi.org/10.5194/acp-2019-1013-AC1). The particles were similar size and grew at the same rate with the regional NPF event particles which would not be expected if it was pollution.

Fig. 8, unprecise timing, missing coordinates, comparison with HYSPLIT GDAS temperature and MBL data suggests a profile at ~ 09:00 instead of 07:15, see also Fig 4. Air time of the Cessna (07:00 – 10:00 UTC?) is questionable.

ANSWER: The GPS coordinates for the Cessna flights can be seen in the next figure. The profile includes data between 7-10 UTC, so there is data from 9 UTC also.

Fig. 9, GPS coordinates, altitudes and time missing. a) It's not clear, how and whether are the two patterns in Fig. 9a from Zeppelin and Cessna are coordinated? Wind direction is also changing during the morning by ~ 20 degree within 2 h (HYSPLIT). Fig.9b) Where and at what altitude has the Zeppelin been between 05:00 and 10:00? 5 hours airborne are ~ 360 km, the loop is only 140 km.

ANSWER: The Cessna and Zeppelin flights took place at the same time. The flight patterns were coordinated to minimize downwind pollution interference.

The wind direction is the mean wind direction during the measurement at the SMEAR II station. The change in wind direction probably affected the movement of the sub-10 nm particle zone over the area (again see Figure 3 in https://doi.org/10.5194/acp-2019-1013-AC1). The purpose of this plot is to show horizontal variation. The loop around Ferrara and the return to the airfield was flown at constant altitude at roughly couple hundred meters above sea level. In the HTL case the zone with higher particle concentration was observed at all altidudes inside the ML.

Fig. 10, air mass back trajectories not matching the position of the Zeppelin. The Zeppelin was airborne that day (June 30) for more than 5 hours (see text) flying at 20 m/sec. The trajectories plotted, however, were obviously not calculated for the time, location and altitude of the actual measurement. The spacing is not in agreement with the operational cruising speed of the mobile platform.

ANSWER: We chose trajectory arrival points in the south and north sectors of the measurement area at 400 m asl altitude and calculated the airmass trajectories arriving to these points at 8:00 UTC. The NPF event started at 8 UTC in SPC, at that time the Zeppelin was also measuring about 5 km north of SPC and observed the starting of the NPF event. Then the Zeppelin began to fly north at roughly 200-300 m asl in order to explore the horizontal extent of the NPF event. Almost immediately when moving north the NPF event was lost. To reach the north point of the flight track took roughly 30 min.

The point of the trajectory analysis was to see if the south sector where the NPF event was taking place had different airmass origin than the north sector. The current analysis confirms this even

though the trajectory end points are not exactly chosen on the flight path. We added information on the trajectories in the caption.

Summary:

A three dimensional airborne study as presented here on a time critical process, 'the onset', requires a clear temporal and geographical identification of data points and a timely correct trajectory analysis. This is unfortunately missing in this manuscript, or, where it is at least partially available, often and obviously false due to incorrect time settings.

ANSWER: The issues presented were addressed. We think the figures are precise enough to support the conclusions.

Finally the abstract claims: 'In Po Valley we observed NPF that was limited to a specific air mass', however, the air mass is neither specified nor characterized in the text. Air mass origin and composition in the text is speculative.

ANSWER: A full analysis with chemical transport models etc. to more completely characterize the airmass is outside the scope of this study.

The reference list is incomplete. Citations are missing for:

Dada et al, 2017, Dunne et al., 2016; Gordon et al., 2017; Yu and Luo, 2009, Mohr et al, 2019, Pierce and Adams, 2009; Stratmann et al, 2003, Junkermann and Hacker, 2018; Sullivan et al, 2016; Vogel and Elbern, 2021

ANSWER: The bibliography has been updated.

However, there are a few more issues that came up reading the manuscript not addressed but likely worth an investigation with a high resolution chemistry transport model:

Interestingly, NPF is linked to convection rather than to air chemistry, how can this be explained? Convection starts with global radiation, air chemistry with UV radiation, respectively photolysis, about 1.5 h later and subsequently it should also take some time for the fresh clusters to grow into measurable particles (Kulmala et al, 2013).

ANSWER: Nilsson et al. (2001) provides some explanations.

What is the origin of the sulfuric acid (SA) at a time of the day (early morning) when there is not yet sufficient UV radiation to produce OH radicals for sulphur dioxide conversion and why is SA observed first in the 'clean' residual layer? What is the corresponding chemistry? The maximum solar radiation flux for June 30 is only ~ 10% different for SPC/Bologna (896) and Monte Cimone (976). Under conditions with more SO2 pollution, in the boundary layer the formation of sulphuric acid (SA) thus should be more intense.

ANSWER: The pseudo SA concentration starts to increase after sunrise, which was at ~4:30 UTC. SO2 concentration is probably low at the surface due to deposition, but remains high in the RL.

Why are nanoparticles not observed in the north-western loop, north of the fiume, despite an advection of an air mass passing over the areas with intense agricultural activities and more intense ammonia emissions (see the Italian emission inventory, Taurino et al, 2020). Contrary, nanoparticles

were observed the same day June 30 in air masses travelling over forested mountains southwest of Bologna. A contribution of the city of Bologna is unlikely based on HYSLIT trajectories for the day as well as a contribution of the Venice area as speculated in the text.

ANSWER: Perhaps in the northern airmass there were also other emissions that prevented the formation of aerosol particles. A more detailed investigation would be needed to fully understand this day.

References:

Nilsson, E. D., Rannik, Ü., Kulmala, M., Buzorius, G., and O'dowd, C. D.: Effects of continental boundary layer evolution, convection, turbulence and entrainment, on aerosol formation, Tellus B, 53, 441–461, https://doi.org/10.1034/j.1600-0889.2001.530409.x, 2001.